# Pilot Study to Detect Genes Involved in DNA Damage and Cancer in Humans: Potential Biomarkers of Exposure to E-Cigarette Aerosols

**DOI:** 10.3390/genes12030448

**Published:** 2021-03-22

**Authors:** Samera H. Hamad, Marielle C. Brinkman, Yi-Hsuan Tsai, Namya Mellouk, Kandice Cross, Ilona Jaspers, Pamela I. Clark, Courtney A. Granville

**Affiliations:** 1UNC Lineberger Comprehensive Cancer Center, University of North Carolina at Chapel Hill, Chapel Hill, NC 27599, USA; ytsai@email.unc.edu; 2Curriculum in Toxicology and Environmental Medicine, University of North Carolina at Chapel Hill, Chapel Hill, NC 27599, USA; ilona_jaspers@med.unc.edu; 3College of Public Health, Ohio State University, Columbus, OH 43210, USA; brinkman.224@osu.edu; 4National Institute of Environmental Health Sciences, Research Triangle Park, NC 27709, USA; namya.mellouk@gmail.com; 5Gad Consulting Services, Risk Assessment, Consulting in Raleigh, Raleigh, NC 27609, USA; kmcross84@gmail.com; 6Department of Pediatrics, School of Medicine, University of North Carolina at Chapel Hill, Chapel Hill, NC 27599, USA; 7School of Public Health, University of Maryland, College Park, MD 20742, USA; 8Drug Information Association, Washington, DC 20036, USA

**Keywords:** DNA damage, gene expression, electronic cigarettes, human buccal and blood, biomarkers, vaping behaviors

## Abstract

There is a paucity of data on how gene expression enables identification of individuals who are at risk of exposure to carcinogens from e-cigarette (e-cig) vaping; and how human vaping behaviors modify these exposures. This pilot study aimed to identify genes regulated from acute exposure to e-cig using RT-qPCR. Three subjects (2M and 1F) made three visits to the lab (n_TOT_ = 9 visits); buccal and blood samples were collected before and immediately after scripted vaping 20 puffs (n_TOT_ = 18 samples); vaping topography data were collected in each session. Subjects used their own e-cig containing 50:50 propylene glycol (PG):vegetable glycerine (VG) +3–6 mg/mL nicotine. The tumor suppressor *TP53* was significantly upregulated in buccal samples. *TP53* expression was puff volume and flow rate dependent in both tissues. In blood, the significant downregulation of N-methylpurine DNA glycosylase (*MPG*), a base excision repair gene, was consistent across all subjects. In addition to DNA repair pathway, cell cycle and cancer pathways were the most enriched pathways in buccal and blood samples, respectively. This pilot study demonstrates that vaping 20 puffs significantly alters expression of *TP53* in human tissues; vaping behavior is an important modifier of this response. A larger study is needed to confirm these relationships.

## 1. Introduction

The use of e-cigarettes (e-cigs) is rising rapidly. In two years (2017–2018), e-cig use increased 77.8% (from 11.7% to 20.8%) among high school students and 48.5% (from 3.3% to 4.9%) among middle school students [1]. Currently, all the major cigarette manufacturers are marketing e-cigs [2]. E-cigs may be less toxic compared to other combustible tobacco products (e.g., cigarettes and cigars) [3]. Some harmful and potentially harmful constituents (HPHCs) that are detected in mainstream smoke of combustible tobacco products were 9–450 times lower in e-cigarette aerosols [4].

However, other carcinogenic HPHCs, including carbonyl compounds, were measured in e-cig aerosols at levels comparable to those in the mainstream smoke of combustible cigarettes [5].

E-cigarettes deliver chemicals through heating e-liquids, which mainly consist of vegetable glycerol (VG) and propylene glycol (PG). The thermal decomposition and oxidation of glycerol and propylene glycol were shown to form low molecular weight carbonyl compounds, including formaldehyde and acetaldehyde [6,7]. The levels of these compounds are highly dependent on the voltage of the e-cig device and the composition of e-liquids [7,8]. These compounds are able to induce DNA adducts [9,10,11], DNA-DNA crosslinks [12], DNA-protein crosslinks (DPCs) [13], and DNA-glutathione crosslinks [14]. The formation of these crosslinks increases cell proliferation leading to mutations which contribute to the carcinogenicity of these compounds [15]. Additionally, genes involved in apoptosis, immunity, metabolism, signal transduction, transportation, coagulation, and proliferation were found to be up- and down-regulated after the exposure to these carcinogens [16]. In addition to aldehyde exposure, e-cigs containing nicotine deliver other carcinogens called tobacco specific nitrosamines (TSNAs) [17]. It has been demonstrated that TSNAs induce DNA damage through DNA adduct formation [17,18], alkyl phosphotriester formation [19], and induction of p53 and Ras mutations [19,20]. 

Both aldehydes and TSNAs are classified by the Agency for Toxic Substances and Disease Registry (ATSDR) as human carcinogens [21]. These carcinogens have been identified as DNA damaging agents, contributing to tumor initiation and development in different tissues [9,22,23,24]. 

DNA damage is usually corrected by the proofreading function of DNA polymerases that can recognize damaged sites in DNA; however, when a cell accumulates a large amount of DNA damage over a short time, its repair systems get saturated and replication occurs in cells with unrepaired lesions, leading to the perpetuation of mutations that can lead to cancer [25]. Efforts have been made to understand the gene expression profile after e-cig exposure [26,27,28]. Although informative, these studies either performed exposure assessment based on in vitro assays or did not account for the endogenous levels of the measured genes in vivo. Additionally, DNA damage resulting from toxicant exposure is dose-dependent [16]. Therefore, it is important to incorporate measures of vaping behavior in studies of e-cig toxicity and exposure. 

This pilot study aimed to identify early molecular events from acute e-cig exposure through DNA damage and repair gene measurements in human buccal and blood samples as a result of scripted e-cig vaping in established e-cig users. Samples were collected before and immediately after vaping so that each subject served as their own control. Thus, any endogenous effects on DNA in buccal and blood samples were controlled for. Associations between human vaping behaviors (puffing topography) and gene expression were also determined. 

## 2. Materials and Methods

An overview of the experimental study design is shown in Appendix A. All samples were collected between 18th May and 5th June 2017.

### 2.1. Study Participants 

This study was performed at the Battelle Memorial Institute/Tobacco Exposure Research Laboratory (TERL) located in Columbus, Ohio. The study was approved by the Battelle Internal Review Board (FWA 0004696). Advertisements in community and college newspapers were used to recruit participants. Interested participants were telephoned to explain the nature of the study, collect basic demographic data (age, gender, ethnicity), and determine the prospective participant’s eligibility to participate in the study (Appendix A). Appendix A shows the telephone survey used to screen for eligibility. Briefly, subjects had to be ≥18 ½ years old, non-smokers or ex-smokers that exclusively used e-cig at least 2 months before participation in the study, and used e-cig at least 8 times a day. In addition, participants had to be healthy; and were excluded from the study if they had significant medical problems including respiratory allergies, a history of pulmonary disease, or asthma. Female participants that were trying to get pregnant, were pregnant or breastfeeding, or not using a reliable birth control method were also excluded (Appendix A). Written consent was obtained from each subject at the start of each visit.

### 2.2. Sample Size

This feasibility study tested the utility of comparing acute measures of toxicity by collecting buccal and blood samples from the same subject before and immediately after e-cigarette use, measuring human vaping behavior during e-cig vaping, and measuring genes that are involved in DNA damage and cancer that could be used as biomarkers of e-cig carcinogenicity. In this pilot study, we recruited 3 subjects, each subject made 3 visits to the lab (total = 9 samples); samples collected before and after exposure to e-cig to established an internal control for each subject and eliminated any endogenous effect on DNA or gene expression. Blood and buccal samples were collected from each subject in each visit (total: 18 samples). No conclusions can be drawn from this small sample size; however, our methods show promise and suggest that the protocol should be used to measure the same genes in a larger sample size/study to confirm gene results and the relationships observed in this pilot.

### 2.3. Human Exposure

Three e-cig users were recruited for in-laboratory scripted use of their own tank style devices. Participants were asked to complete three separate visits. During each visit, subjects were asked to vape 20 puffs, according to scripted vaping (three seconds puff every 60 seconds, for a total of 20 puffs over 20 min). To execute the scripted vaping, participants watched a computer screen that told them when to inhale and when to exhale.

Before vaping, a Vitalograph Breath CO Monitor was used to measure CO levels in exhaled breath, and blood and buccal epithelial samples were collected. Immediately after vaping 20 puffs, exhaled breath CO was measured, and blood and buccal samples were collected. Participant processing dates, e-cig devices, and exhaled CO levels are shown in Appendix A; blood and buccal samples collected are outlined in Appendix A.

### 2.4. Topography Data Collection

The user’s e-cig was connected to a smoking puff analyzer device (SPA-D, Sodim, FR) which recorded subject-specific topography, including puff volume, puff duration, flow rate, interpuff interval, pressure drop, and average resistance during vaping.

### 2.5. Biologic Sample Collection

For buccal sampling, a cytology brush (Puritan^®^ 2188 Histobrush) Cytology Brushes with Soft Tapered Tip, Nylon Bristles & Polypropylene Shaft) was used to collect buccal samples from the inner cheek before and after vaping. After buccal sampling, the cytology brush was immediately immersed in a RNAlater solution and removed, the sample was vortexed, and kept frozen (at −80 °C) until analysis. Participant duplicates were processed the same way as the samples (Appendix A). Method blanks were prepared by immersing the cytology brush in a RNAlater solution and processed the same way as the sample. Including duplicates and blanks, from three subjects and three visits, a total of 25 buccal samples were processed (Appendix A).

Sodium heparin mononuclear cell preparation tubes (CPT) were used to collect blood samples before and after vaping, for gene expression analysis. After sampling, tubes were left for 5 minutes before centrifuging at ~4000 rpm for 15 min at room temperature for white blood cells (WBCs/buffy coat) collection. The buffy coat was then aspirated and added to RNAlater solution, vortexed, and kept frozen (at −80 °C) until analysis. Blanks were prepared using an RNAlater tube and processed the same way as the samples. A total of 25 blood and 25 buccal samples, including duplicates and blanks (Appendix A), were shipped to the Battelle Eastern Science and Technology (BEST) for gene expression analysis. 

### 2.6. Quantitative RT-PCR Analysis of Gene Expression

Details on gene expression analysis are available in the Appendix A. Briefly, fifty biologic (25 buccal epithelial and 25 blood) samples that were stored in RNALater were centrifuged at 5000 rpm 4 °C for 10 min to separate the cells from the solution. The RNA extraction and purification were performed using the RNeasy Mini Kit. cDNA was synthesized using SABiosciences RT^2^ First Strand Kit, which was then combined with SYBR Green qPCR Mastermix and distributed across the array (SABiosciences/Qiagen, Alameda, CA, USA Cat#: PARN-11C). The genes featured in this kit are those associated with the transcriptional targets of DNA damage responses (Appendix A). Each array contains appropriate housekeeping genes, genomic DNA amplification, reverse transcriptase, and positive PCR controls (see Appendix A). The catalog numbers of the RT-PCR primers used in this study, are presented in Appendix A. The data were analyzed by an integrated Excel-based template that automatically performed ΔΔCt-based calculations from the raw threshold cycle (Ct_cutoff_ = 35) value to determine gene-specific expression. The Excel template normalized the Ct values of samples (collected after e-cig exposure) with the control (samples collected before the exposure to e-cig) values to calculate and report the fold-change of exposed samples compared to control. 

### 2.7. Gene Network Analysis and Biomarker Identification

A core analysis, including a downstream effects analysis, was performed using the Ingenuity Pathway Analysis Propriety Tool (IPA; http://www.ingenuity.com (accessed on 16 October 2020), Redwood City, CA). Downstream analysis was used to identify biological processes along with gene network based on changes in expression of given genes. The calculation of the overlap P-value (*p*) and the activation Z-score was based on Fischer’s exact test. We applied the following cut off: −1.5 > fold change > 1.5, and *p* < 0.05. 

### 2.8. Statistical Analysis

Although the number of samples measured for gene expression was relatively small, statistical analysis employed on buccal (n = 9) and blood (n = 9) samples identified genes that are differentially expressed relative to samples before exposure (controls). This controls for and essentially excludes any endogenous effects from the assessment of e-cig exposure-based gene regulation. The fold-change of the exposed samples compared to controls was used to identify differentially expressed genes (DEGs). To test if a gene is differentially expressed, we performed a one sample t-test of the log2(fold-change) of all samples to determine whether the sample mean is statistically different from zero, which means no change.

For association between gene expression and vaping behaviors (e.g., puff volume, duration, etc.), a two sided Pearson correlation test (cor.test(method = “pearson”, alternative = “two.sided”) in R) was performed. Any gene with a *p*-value less than 0.05 was considered to be significantly correlated with the vaping behavior tested.

GraphPad Prism 8 software (GraphPad Software, San Diego, CA, USA) was used to present the data.

## 3. Results 

### 3.1. Genes Related to DNA Damage Are Differentially Regulated in Buccal and Blood Samples after E-Cig Exposure 

Eighty-four genes related to DNA damage were analyzed to understand the relationship between e-cig exposure and DNA damage (the names of genes that are included in this study are presented in Appendix A). Using the criteria of fold change ≥ +1.5 or < 0.66 (1.5-fold change downregulation) in mean gene expression of all subjects in all visits, 61 genes were differentially expressed in buccal samples, and 70 genes were differentially expressed in blood samples (Figure 1(A1,A2)). In order for the gene to be significantly expressed, it should pass the *t*-test of the log2(fold-change) of all samples as a mean which should be statistically different from zero. Of these genes, five were significantly (*p* < 0.05) upregulated in buccal samples including flap structure-specific endonuclease 1 (*FEN1*), apoptosis inducing factor mitochondria associated 1 (*AIFM1*), X-ray repair cross complementing 2 (*XRCC2*), three prime repair exonuclease 1(*TREX1*), and tumor suppressor *TP53* gene (Figure 1(B1)). In blood samples, only N-methylpurine DNA glycosylase (*MPG*) repair gene was significantly (*p* < 0.05) downregulated (Figure 1(B2)).

The DEGs were also classified using the ingenuity pathway analysis (IPA) to examine the biological pathways and the associated functions that are triggered after e-cig exposure (Figure 2). In both buccal and blood samples, the DNA replication, recombination, and repair pathway was the major pathway activated by e-cig exposure (Figure 2A,B). However, this pathway includes a wide range of sub-pathways, including mismatch DNA repair (MMR), nucleotide excision repair (NER), base excision repair (BER), homologous recombination (HR), and non-homologous end joining (NHEJ) [29]. The DEGs that are involved in the DNA repair pathway were further visualized by network analysis (Figure 2A,B). This tool revealed the coordinated up and down regulation of genes associated with DNA damage and repair that occurred in both tissues. For example, the tumor suppressor *TP53* gene was significantly upregulated (Figure 2A) in buccal samples (Figure 1(B1)). In blood samples, *TP53* regulation was attenuated (Figure 2B).

Network analysis also showed changes in expression of other genes that are associated with DNA damage and repair after e-cig exposure, most of these genes were not significantly expressed by subjects mean, except the genes presented in Figure 1(B1,B2). The general transcription factor IIH subunit 1 (*GTF2H1*) along with RAD51 paralog B (*RAD51B*), X-ray repair cross complementing 3 (XRCC3), and three prime repair endonuclease 1 (*TREX1*), the dsDNA repair factors, were shown to be upregulated in blood (Figure 2B). On the other hand, N-methylpurine DNA glycosylase (*MPG*), the BER gene, was extremely downregulated (Figure 2B). The same network analysis of buccal samples revealed downregulation of excision repair-1 endonuclease non-catalytic (*ERCC1*) gene, mismatch- MutS homolog 3 (*MSH3*) repair gene, and double strand DNA (dsDNA) break- X-ray repair cross complementing 6 (*XRCC6*) repair gene (Figure 2A). 

Other signaling pathways were also activated by e-cig vaping. In blood, cancer appears to be the second most enriched pathway after DNA replication, recombination, and repair (Figure 2B). In buccal epithelium, while the cancer pathway appeared among the top 15 triggered pathways after e-cig exposure, cell cycle was the second most enriched pathway after e-cig exposure (Figure 2A). Figure 3 shows that several DEGs contribute to both DNA repair and cancer; 15 genes in buccal and 43 genes in blood samples (Figure 3A,B). Additionally, the p38 mitogen-activated protein 2 kinase-6 (*MAP2K6* gene; also known as *MKK6*) cancer gene was differentially expressed in the buccal and blood samples of all subjects after e-cig exposure (Figure 3A,B). Further, the fold change of *MAP2K6* gene in buccal samples was consistent (positively correlated) with its level in blood samples of each subject (S01: R^2^ = 0.963; S02: R^2^ = 0.806; and S03: R^2^ = 0.649), figures are not presented. Three other cancer genes including mitogen-activated protein kinase 12 (*MAPK12*), cell death inducing *DFFA* like effector A (*CIDEA*), and inositol hexa-kis-phosphate kinase 3 (*IP6K3*) were upregulated after e-cig exposure in blood, but not in buccal samples (Figure 3A,B).

### 3.2. Significant Differential Expression of Genes Is Consistent among Participants

The significantly (*p* < 0.05) and differentially expressed genes in buccal and blood samples of all subjects are presented (by subject) in Figure 4. The *MPG* gene was significantly downregulated in blood (Figure 1(B2)); this was comparable across all subjects (Figure 4); but the differential expression of *MPG* in buccal samples was incomparable (Figure 4). However, the higher *MPG* expression appeared to be accompanied by the lower *TP53* expression in buccal samples (Figure 4). In buccal tissue, while the upregulation of *FEN1*, *TREX1*, and *XRCC2* was consistent across all subjects, *AIFM1* and the tumor suppressor *TP53* were different across the three subjects (Figure 4). In blood, differential expression of all five genes was consistent between visits, for subjects 1 and 2, but not subject 3 (S03) where greater variation in gene expression was observed (Figure 4). Differential expression of these genes was not significant in blood (Figure 1(B2) and Figure 4).

In order to test whether the correlation of *MPG* and *TP53* was significant, we performed a regression line of *MPG* versus *TP53* in buccal samples. Figure 5A shows strong negative correlation by subject (mean average of all visits per subject; R^2^ = 0.880). However, this correlation was reversed in blood samples where a positive correlation was observed between *MPG* and *TP53* expression (mean average of all visits per subject; R^2^ = 0.608) (Figure 5B).

### 3.3. Gene Regulation Is Vaping Behavior-Dependent

In order to understand the effect of vaping behaviors on the DEGs, we presented the DEGs that are associated with vaping behaviors by visit (Figure 6). Average puff volume was associated with changes in expression of several genes, especially *TP53* in both buccal and blood samples (Figure 6A,B). The greater expression of several genes was associated with the greater puff volume and flow rate (Figure 6B). In addition to puff volume and flow rate, *TP53* levels in buccal samples appeared to also be sensitive to pressure-drop and resistance, where the lower pressure-drop and resistance were associated with the higher expression of tumor suppressor *TP53* (Figure 6A). Overall, flow rate was associated with expression of a very few genes in buccal epithelium. In blood, changes in flow rate were associated with changes in expression of several genes, including *MAP2K6*. *MAP2K6* showed higher expression in buccal and blood samples when puff volume was larger (Figure 6A,B). Larger puff volume was also associated with increased differential expression of 8-oxoguanine glycosylase 1 (*OGG1*; a lung cancer [30]) gene in buccal and blood samples (Figure 6A,B). 

## 4. Discussion

E-cigarettes are thought to be safer than conventional cigarettes; however, very little has been documented with respect to understanding their toxicity to humans. DNA damage is one of the major concerns of exposure to carcinogens from tobacco products. After exposure to toxicants, cells adapt several pathways to repair DNA damage through the activation of genes involved in these pathways [29]. However, these early molecular events could contribute to developing hyperplastic, metaplastic, and neoplastic lesions if they go unrepaired. 

This pilot study aimed to identify the early molecular events that are initiated after acute exposure to e-cigarette in humans. Identifying these early molecular events might enable developing strategies to mitigate the toxic effect of tobacco products. Additionally, individual response and susceptibility to these exposures are influenced by several factors including the genetic background, co-exposures to environmental agents, age, and sex. Therefore, an individual control is critical to accurately measure the significantly altered genes after exposure. 

This study showed that even short-term e-cigarette exposure impacts expression of several genes associated with DNA damage, DNA repair, cell cycle, and cancer. Genes related to DNA damage were differentially (1.5-fold change up or down) regulated in both buccal and blood samples, immediately after vaping 20 puffs using e-cigs with (50:50) PG/VG + 3–6 mg/mL nicotine. In buccal samples, the *TP53* tumor suppressor gene was significantly upregulated (Figure 1B). This is important because in normal cells, *TP53* is expressed at very low levels. However, upon exposure to toxicants, *TP53* levels increase, leading to cell cycle arrest at the checkpoint until the DNA damage is repaired [31]. It is not surprising that, in parallel with the *TP53* upregulation in buccal, *TP53*-dependent genotoxic stress inducible genes that are involved in regulation of cell cycle, proliferation, and differentiation are upregulated as well (i.e., *AIFM1*, *FEN1,* and *TREX1*) [32,33]. It is well known that *TP53* modulates the transcription of a vast number of genes that are involved in cell cycle control, apoptosis, differentiation, and DNA repair [34,35]; however, different genes respond differently to different levels of *TP53*, based on the type of DNA error that occurs after exposure [36]. Our results show that the cell cycle is the second most enriched pathway triggered by e-cig vaping, after DNA damage pathway, in buccal samples (Figure 3A), which is consistent with the upregulation of *TP53*, a major cell cycle and tumor suppressor gene [31]. 

The role of *MPG* in the BER pathway is to recognize and excise damaged nucleotides but, under certain conditions, *MPG* contributes to the formation of sister chromatid exchange and chromosomal aberration [37,38]. Additionally, *MPG* was found to be upregulated in several types of cancers [38,39,40,41]; it interacts with and inhibits the tumor suppressor *TP53* protein and its target genes [42]. Our results demonstrated an inverse association between *MPG* and *TP53* expression in buccal epithelium (Figure 5A; R^2^ = 0.880). However, the downregulation of *MPG* in blood cannot be explained by this model (Figure 5B). Larger (in vitro and in vivo) studies might clarify the correlation/interaction between *MPG* and *TP53* genes and the effect on *TP53* downstream targets in different tissues, after e-cig vaping.

*MAP2K6*, a cancer-related gene, was consistently upregulated in both buccal and blood samples following vaping. Although its expression was not significant in this small study, it might be used as a marker if confirmed by a larger study.

The observed impact of vaping behavior on gene levels is not surprising given that studies in combustible products have demonstrated that smoking behavior impacts exposure [43]. In this study, while puff volume was associated with changes in expression of a large number of genes in both buccal and blood samples, flow rate showed effects on several genes in blood but not in buccal samples (Figure 6A,B). The tumor suppressor *TP53* seemed to be sensitive to puff volume and flow rate in both tissues (Figure 6A,B). Our results show that vaping behavior impacts gene expression, thus, e-cigarette toxicity. However, a larger sample size is needed to substantiate the findings observed here and to further our understanding on how different vaping behaviors could cause aberrant gene regulation and impact e-cigarette toxicity.

## 5. Conclusions

This pilot study shows preliminary evidence that vaping 20 puffs of an e-cigarette is sufficient to cause significant changes in expression of the tumor suppressor *TP53*, in addition to other cancer-related genes (e.g., *MPG*) in humans. These early molecular events could be used as biomarkers of exposure to carcinogens from e-cigarettes; however, larger follow-up (in vitro and in vivo) studies are needed to understand how these molecular events contribute to the accumulation of unrepaired lesions. This study also showed corradiated expression of *MPG* and *TP53* following vaping, and that vaping behaviors impact gene expression changes in relevant target tissues. These preliminary results require confirmation; and a larger study in humans is needed to better understand the impact of e-cig use on these genes and pathways, as well as how vaping behavior modifies e-cig exposure and toxicity.

## Figures and Tables

**Figure 1 genes-12-00448-f001:**
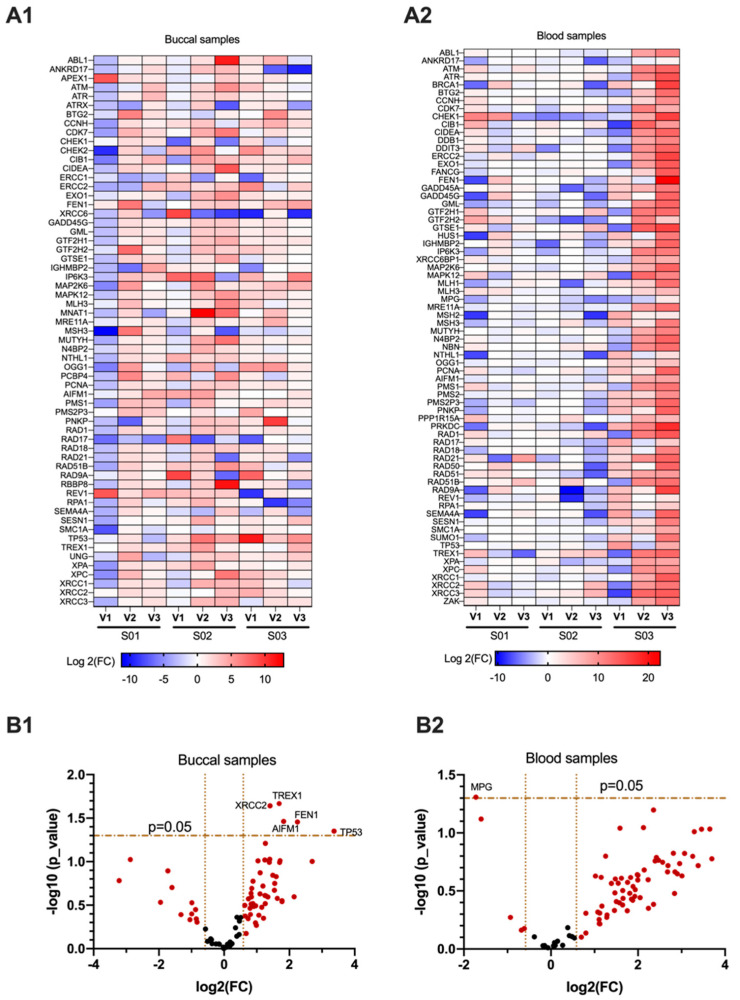
Differentially (1.5-fold change up or down) regulated genes in buccal and blood samples for three subjects across three visits. (**A1**,**A2**) Heat maps showing log2 fold change (FC) after vaping; almost all DNA damage genes were differentially upregulated in blood samples of subject 3 (**A2**), compared to subjects 1 and 2. (**B1**,**B2**) Volcano plots showing the significantly (*p* < 0.05) expressed genes after e-cig exposure. Five DNA damage genes including tumor suppressor *TP53* were significantly upregulated by mean of all buccal samples (**B1**), while N-methylpurine DNA glycosylase (*MPG*), was the only gene that significantly downregulated by mean of all blood samples (**B2**).

**Figure 2 genes-12-00448-f002:**
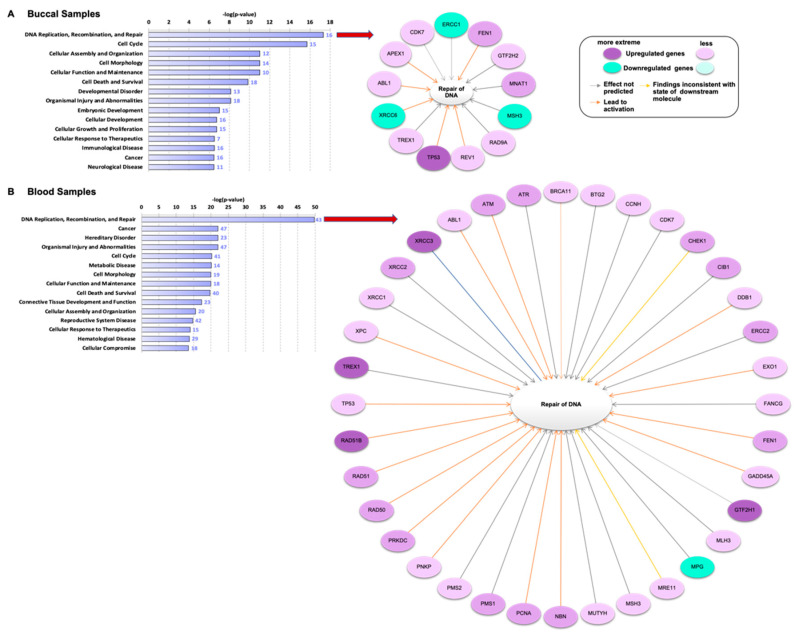
Top 15 significantly (*p* < 0.05) pathways triggered after e-cig exposure. (**A**) Results of ingenuity pathway analysis showing the most significant biological functions in buccal and (**B**) blood samples, following vaping for three subjects across the three visits. The numbers at the end of each bar represent the number of genes involved in the pathway. Genes associated with DNA repair pathway are presented in the network analysis. Cell cycle was the second most significant pathway activated after e-cig exposure in buccal epithelium, while in blood, cancer was the second most significantly triggered pathway following e-cig exposure.

**Figure 3 genes-12-00448-f003:**
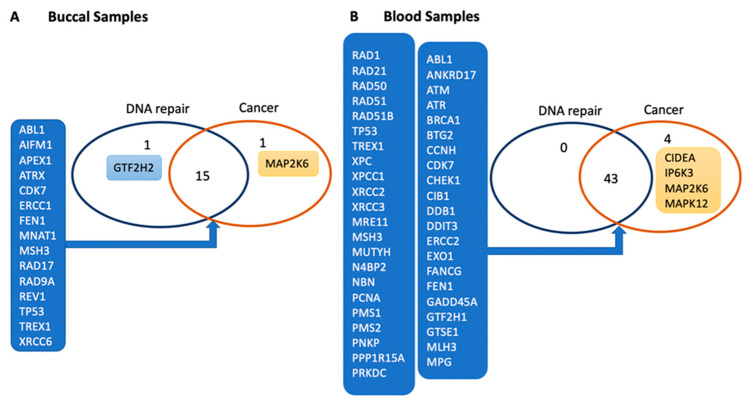
Venn diagram of genes involved in DNA repair and cancer in buccal and/or blood samples. (**A**) Fifteen genes that are differentially expressed in buccal epithelium and 43 genes that are differentially expressed in blood are involved in DNA repair and cancer (**B**). Mitogen-activated protein 2 kinase-6 (*MAP2K6*) cancer gene is differentially expressed in buccal epithelium and blood samples (**A**,**B**). Four cancer genes, including *MAP2K6,* were differentially (but not significantly) expressed in blood samples (**B**).

**Figure 4 genes-12-00448-f004:**
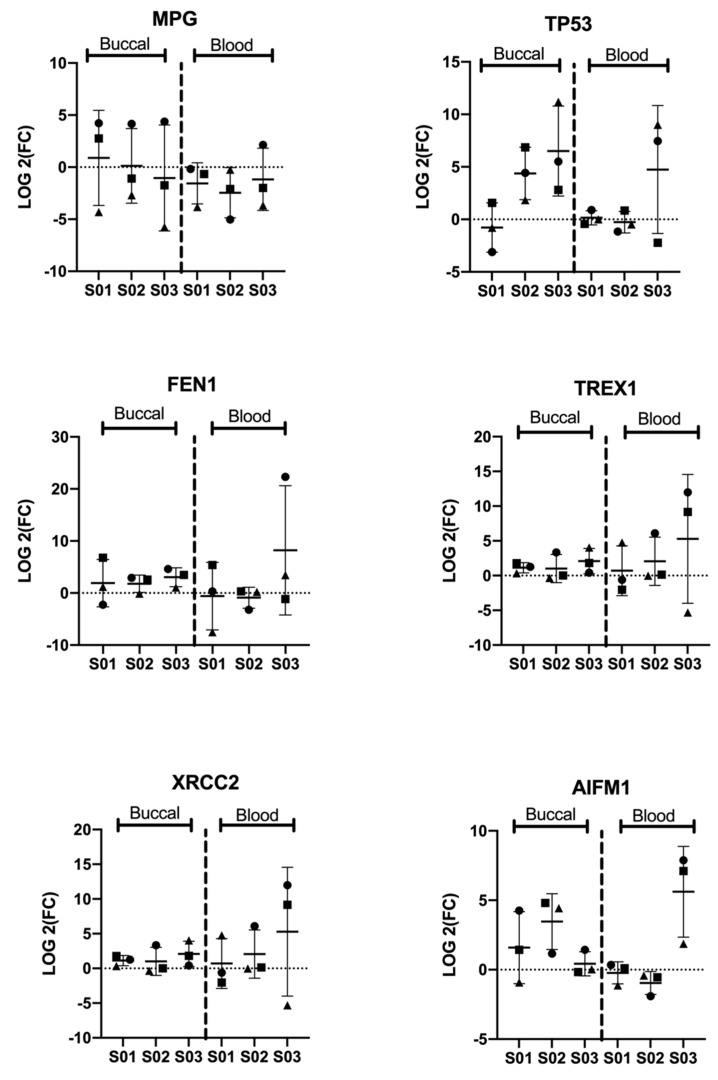
Log2 fold change (FC) of the significantly (*p* < 0.05) and differentially (1.5-fold change up or down) regulated genes in buccal and blood samples across the three visits (V1-V3; n_TOT_ = 9). Triangle refers to visit 1; square to visit 2; and ball to visit 3. N-methylpurine DNA glycosylase (*MPG*) was significantly downregulated in blood samples; other genes, including tumor suppressor *TP53*, were significantly upregulated in buccal epithelial samples.

**Figure 5 genes-12-00448-f005:**
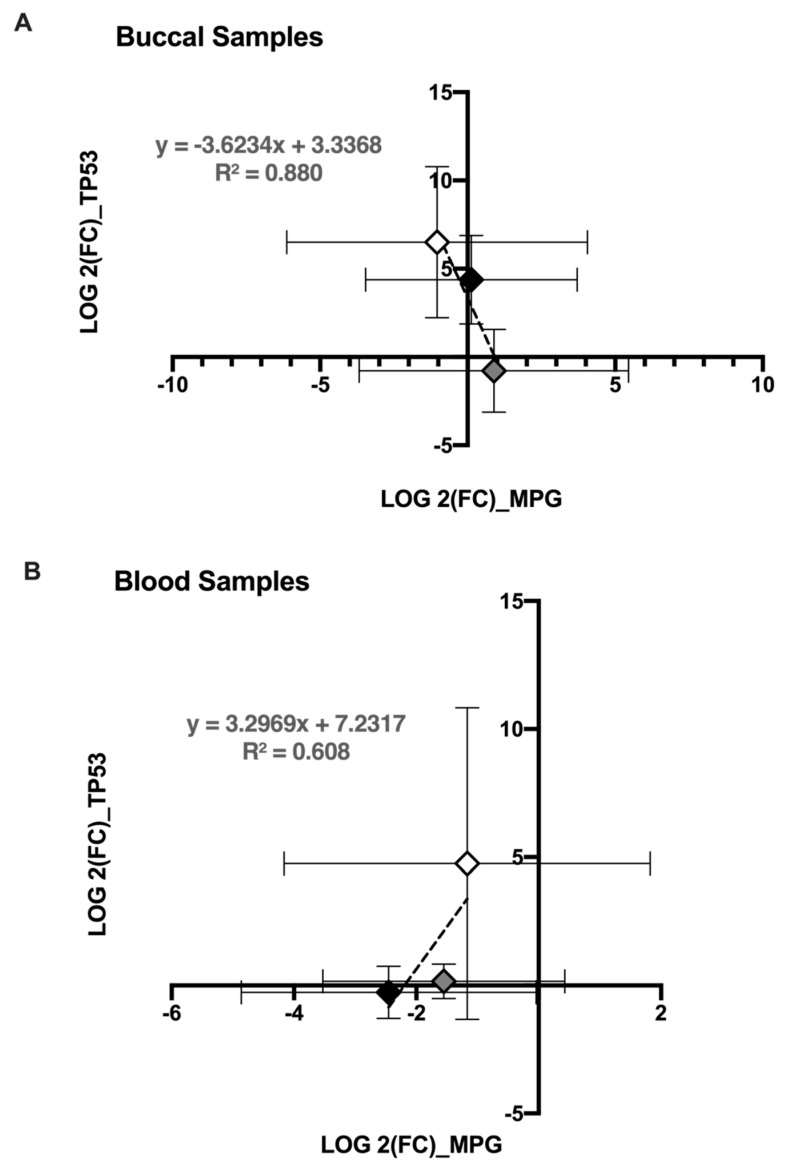
*MPG* expression is correlated with *TP53*. Expression of *MPG* and *TP53* are negatively correlated in buccal epithelium (**A**) and positively correlated in blood (**B**) following vaping; each point represents the average of log 2(FC) of the gene in the three visits for each subject; gray diamond refers to subject 1; black diamond to subject 2; and white diamond to subject 3. Error bars represent the standard errors of the log 2(FC) of gene expression for the 3 visits.

**Figure 6 genes-12-00448-f006:**
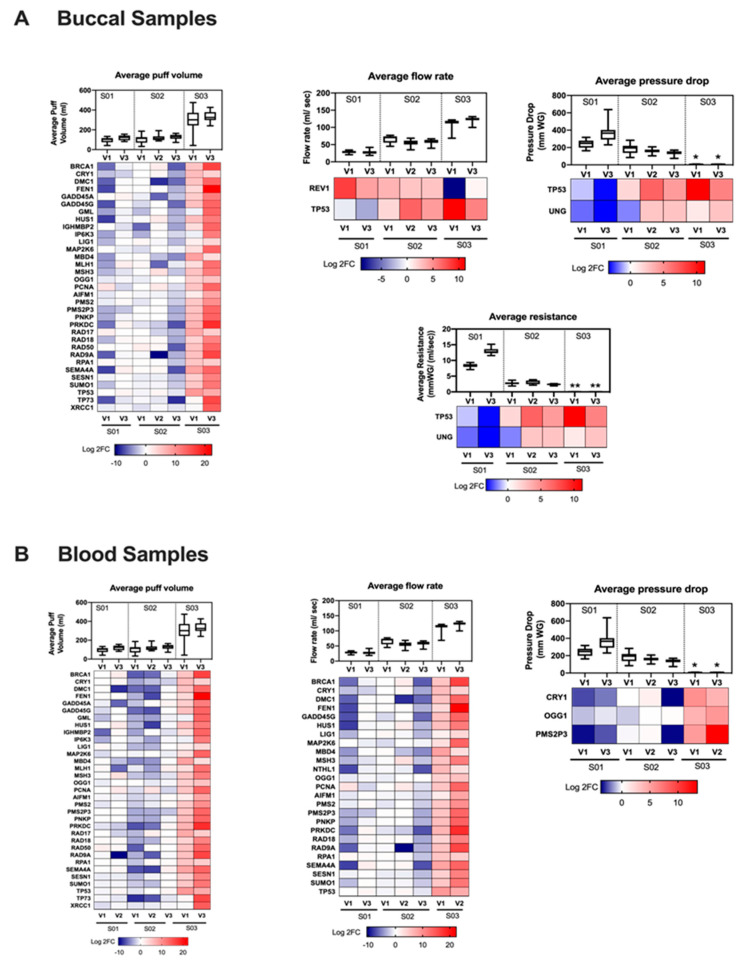
Genes regulated in buccal and blood samples, are vaping behaviors-dependent. Heat maps of genes that have the same trend of vaping behaviors of all subjects across the three visits in buccal (**A**) and blood (**B**) samples (n = 9) along with boxplots of clinical variables (n = 20 puffs); *: Average pressure drop of Subject 3 in visit 1 = 6.8 ± 1.5 and visit 3 = 7.3 ± 1.1 mmWG; **: Average resistance of Subject 3 in visit 1 = 0.06 ± 0.01; and visit 3 = 0.05 ± 0.008 mmWG/(mL/s). The expression of tumor suppressor *TP53* is puff volume and flow rate dependent in both tissues (**A**,**B**).

## Data Availability

The original contributions presented in the study are included in the Appendix A. Further inquiries can be directed to the corresponding author/s.

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
