# Peer review of "Pilot Study to Detect Genes Involved in DNA Damage and Cancer in Humans: Potential Biomarkers of Exposure to E-Cigarette Aerosols"

_genes, 2021, doi:10.3390/genes12030448_

Round 1
Reviewer 1 Report
reviewGenes
Hamad et al. study the ffect of exposure to e-cigarette smoke on the expression of genes invoved in cancer. In 9 individuals, who were analyzed several times before and after exposure, qPCR showed differential expression in several genes linked to DNA repair, cell cycle and cancer.
The paper is well written and the results are clearly represented. As the authors clearly indicate, this is a pilot study that needs confirmation by a larger study. Since I am not a specialist on cancer or cell cycle, I focused on the data analysis. There are no real errors or flaws in the analysis, but some improvements are possible.
Minor comment
Three individuals were included in the study, who were each analyzed three times. Durin each of these 3 visits, a pre- and a post-exposure measurement was carried out. This had the advantage that each individual is it’s own control. However, these are not 9 independent observartions. Whenever a p-value is calculated, the non-independence between observations within the same individual should be taken into account. This is feasible by a linear mixed model, where individual ID is entered as a random intercept term, plus a random slope for the interaction time*ID. P-values for the effect of vaping are then obtained by the significance of the fixed effect of time (pre- or post-exposure). These p-values are more correct than the p-values obtained using the one-sample t-test described in the manuscript.
In addition, this linear mixed model allows to obtain an estimate of the fraction of the variance in outcome between individuals. More in particular, the variance in the random slope shows how consistent the difference between the pre- and the post- expression outcome is between individuals. Is the variance in random slope accounts for a very small fraction of the variance, the difference pre-post is consistent between individuals.
The p-values in Figure 1 are not corrected for multiple testing. Since this is a pilot study, is understandable that the small sample size does not lead to very low p-values. I would recommend adding an false discovery rate (FDR) analysis, to evaluate if the observed p-values distribution is enriched in low p-values. This can be done using a q-value calculation, or by plottting the observed q-values versus the uniform distribution U(0,1), which is the expected distribution under the null hypothesis that none of the genes is significantly upregulated.
Author Response
Dear Reviewer,
Thank you vey much for your comments and constructive criticism to improve the manuscript. Attached are the revised manuscript and our detailed responses to your comments.
Thank you for your assistance in the review of the subject manuscript.
Dr. Samera Hamad
Lineberger Comprehensive Cancer Center
University of North Carolina at Chapel Hill

Reviewer 2 Report
The manuscript entiteled "Pilot study to detect genes involved in DNA damage and cancer in humans: Potential biomarkers of exposure to E-Cigarette aerosols" by Hamad S.H. et al. describes the effect of smoking e-cigarettes on the expression of genes mainly involved in DNA repair and carcinogenesis. The authors presented a pilot study on 3 volunteers from whom blood and buccal samples were collected at 3 time points before or after exposure to e-cigarettes.
The undoubted disadvantage of the study is the small number of participants. Moore CG et al. in Recommendations for Planning Pilot Studies in Clinical and Translational Research, recomend at least 12 participants for pilot studies (doi: 10.1111/j.1752-8062.2011.00347.x) Considering that the samples were collected in 2017, and the number of e-cigarette smokers increases every year, as the authors also mention in the introduction to the paper, it is surprising that this group has not been enlarged so far.
Nevertheless, taking into account the promising results of the work, as well as the growing number of smokers, especially among teenagers, in addition to increasing the number of participants, it requires major revisions:
- The study involved smoking volunteers, from whom biological material was collected and gene expression studies were performed. However, there is no information about the approval of the study by the Ethics Committee.
- The study should include information on why the studied group was so small
- Revisions in the introduction:
- in references 3 and 4 we can read that e-cigarettes are less toxic, but this is the opinion of the respondents, not the test result, therefore the sentence: "E-cigs may be less toxic compared to other combustible tobacco products (e.g., cigarettes and cigars)" (lines 42-43) may be misleading and the sentence should be corrected or references changed
- reference 5 is for exhaled smoke and in reference 6 there is no information about e-cigarettes, please correct the sentence in lines 43-46
- in verses 46-48 the authors write: "But other carcinogenic HPHCs, including carbonyl compounds, were measured in e-cig aerosols at levels comparable to those in the mainstream smoke of combustible cigarettes", however, reference 7 reports that levels of carcinogenic HPHCs were significantly lower in e-cigarettes
- Revisions in Materials and Methods:
- supplementary figure 1: the drawing should be divided into e.g. A (upper) and B (lower) parts, a more detailed description of the drawing is necessary, the background must be brighter and simpler, no figures of a human body fragment are needed
- supplementary table 2: please describe what "DK" means
- Revisions in Results:
- Figure 1: divided into e.g. 1A 2A / 1B 2B or A (right, left)
- Figure 2: correct the figure so that the gene names are in one line, there is also no difference in "Downregulated genes: more / less" in the top description of the figure
- line 245: no information provided if the changes were statistically significant
- line 252: what does "differentially" mean? this term does not follow from either Figure 1, 3 or 6
- line 254-261: the order of the described genes should follow the order of genes in figure 4
- line 259: gene expression was comparable, not "consistent"
- verse 268: what does "significantly" mean
- line 270-271: the sentence does not follow from figure 4
- line 274-276: R2 = 0.608 and error bars are large, so it cannot be said that "the correlation was significantly reversed"
- Figure 5: "circle", "triangle", "square" here mean subjects, and in previous figures - visits. Need to be standardized
- verse 287: "is clear" - please explain
- Revisions in in the discussion
- line 339-341: reference 39 does not mention MPG
- line 341-342: reference 41 does not mention MPG, more examples are needed
Author Response
Dear Reviewer,
Thank you very much for your comments and constructive criticism to improve the manuscript. Attached are the revised manuscript and our detailed responses to the your comments.
Thank you!
Dr. Samera Hamad
Lineberger Comprehensive Cancer Center
University of North Carolina at Chapel Hill

Round 2
Reviewer 2 Report
I have no further comments. I recommend acceptance.